# The Behavioral and Productive Characteristics of Japanese Quails (*Coturnix japonica*) Exposed to Different Monochromatic Lighting

**DOI:** 10.3390/ani14030482

**Published:** 2024-02-01

**Authors:** Sezgi Karal, Firdevs Korkmaz Turgud, Doğan Narinç, Ali Aygun

**Affiliations:** 1Department of Animal Science, Faculty of Agriculture, Akdeniz University, 07070 Antalya, Turkey; memayt@yandex.com (S.K.); firdevskorkmaz@akdeniz.edu.tr (F.K.T.); 2Department of Animal Science, Faculty of Agriculture, Selçuk University, 42130 Konya, Turkey; aaygun@selcuk.edu.tr

**Keywords:** lighting, growth curve, tonic immobility, bilateral symmetry, behavior

## Abstract

**Simple Summary:**

Light wavelength is very important for poultry physiology. In this study, five monochromatic lightings (white, green, yellow, blue, red) were applied to Japanese quails during the fattening period. The best results in terms of growth, viability, welfare, and developmental stability were found in quails treated with blue and green monochromatic lightings.

**Abstract:**

The purpose of this study is to examine the impact of monochromatic illuminations at wavelengths of white (400–700 nm), green (560 nm), yellow (580 nm), blue (480 nm), and red (660 nm) on the performance and behavioral traits of Japanese quails throughout their fattening period. A total of 300 quails in five lighting experimental groups were housed in their conventional rearing cages. Weekly live weights of quails were measured individually, developmental stability was determined, and behavior and fear tests were performed. The body weight averages of quails exposed to blue, green, and yellow light were determined to be greater than those exposed to white and red light (*p* < 0.05). In terms of the mature weight parameter and the weight of the inflection point of the Gompertz growth model, the averages of the quails exposed to green and blue monochromatic lighting were higher (*p* < 0.05). The most negative findings on aggressive behavior were observed in birds exposed to monochromatic yellow and red light. Although the body weight of quails exposed to yellow monochromatic lighting was similar to that of quails exposed to green and blue monochromatic lighting, green and blue monochromatic lighting produced the best results in terms of growth, behavior, and developmental stability characteristics. Consequently, it is believed that consistently using green or blue monochromatic lighting programs when raising Japanese quails may provide economic advantages to the producers.

## 1. Introduction

Lighting is one of the most important environmental aspects for poultry reared for meat and egg production. Photoperiod and light intensity have long been accepted as major topics of lighting studies for poultry [1]. In addition, numerous recent studies have evaluated the effect of the type of light source that birds are exposed to. With the discovery of new lighting technologies, light wavelength has recently attracted the attention of numerous researchers [2,3,4,5].

There are four types of photoreceptors with maximum sensitivity to light wavelengths of 415, 450, 550, and 700 nanometers (nm) in the eyes of chicken [6]. The bird’s tetrachromatic vision is attributed to these four kinds of retinal photoreceptors [7]. Avian species possess extraretinal photoreceptors, in addition to their natural ones. Circadian rhythm, endocrine functions, avian locomotion, and reproduction are all regulated by extraretinal photoreceptors located in the pineal gland and hypothalamus [8]. Due to this heightened visual sensitivity, certain sources of light are perceived more intensely by birds than by humans. Important physiological and behavioral responses ensue as a consequence of this circumstance [9].

Some researchers have noted that as the wavelength of light approaches ultraviolet, the growth and development characteristics of chicken improve, and their temperament becomes more tranquil [10,11,12]. Halevy et al. [13] suggested that the excitation wavelength of blue and green light causes muscular satellite cells to proliferate more rapidly, thus accelerating the growth of poultry. Contrary to this, as the wavelength of the light approaches infrared, improvements in features such as reproduction and egg production are observed, and animals become more active [14,15]. However, there are also studies reporting findings to the contrary or claiming that the wavelength of light does not affect yield or behavioral characteristics in poultry [16,17,18,19,20].

In the majority of studies examining the impact of different wavelengths, chicken was employed, while very few studies involved other poultry species. The Japanese quail, the smallest poultry species commercially raised for valuable meat and eggs, stands out for its disease resistance, early and rapid development, appropriateness for small farm production, short intergenerational duration, and remarkably high productivity [21,22]. It is also considered a model animal since the results of scientific studies conducted with Japanese quail can be applied to other poultry species. In general, commercial quail producers are small scale, and their flocks are not genetically improved, resulting in low productivity [23]. In circumstances where genetic improvement of the genotype is not achievable according to the general principles of animal breeding, it is well established that environmental modifications will have a favorable influence on productivity. Therefore, the effects of environmental modifications on quail production are an intriguing topic. The objective of this research is to investigate the impact of monochromatic lighting (white, green, yellow, blue, and red) with specific wavelengths (400–700 nm, 560 nm, 580 nm, 480 nm, and 660 nm) on the overall performance and behavioral traits of Japanese quails over all of their fattening period.

## 2. Materials and Methods

All aspects pertaining to the care and administration of the quail subjects utilized in this research were conducted in adherence to the relevant laws and regulations of the Republic of Turkey. The research was carried out at the Animal Husbandry Facilities of the Department of Animal Science at the Faculty of Agriculture at Akdeniz University (Turkey). Animal material for the study consisted of 300 quail chicks obtained concurrently from a randomly mated, previously unselected parent flock.

In order to establish the study’s treatment groups, two replications of each of the five lighting trials were created. These experimental groups were exposed to blue (480 nm), white (400–700 nm), green (560 nm), yellow (580 nm), and red (660 nm) illumination during the growth period in Japanese quails. Wing numbers were assigned to 60 one-day-old chicks, randomly assigned to each experimental group, and placed in rearing cages. The individual body weights of the quails were weighed weekly from hatching to the 42nd day of slaughtering age. The chicks were housed in dedicated brooder cages in the darkened experimental rooms, with a placement frequency of 75 cm^2^/quail from hatching until the third week of sex determination. In brooder cages, 32 °C and 55% RH were provided; the temperature was reduced by 1 °C every three days, and at the end of the second week, the temperature was maintained at 27 °C. Throughout the experiment, the quails were fed ad libitum a diet consisting of 24% crude protein and 2900 kcal/kg metabolizable energy [24]. Following the third week, the chicks were moved to five-story fattening cages in the same rooms, with three compartments on each level. Furthermore, they were kept in cages with a stocking density of 220 cm^2^/quail until the end of the study. On each cage floor, LED lighting devices (Mervesan, one chip with 2700 K, Turkey) were installed, and it was ensured that there was no reflection in the room. Suitable LED lamps were utilized to produce 15 lux of illumination at bird height with continuous lighting.

The growth of the quails was examined using the following form of the three-parameter Gompertz nonlinear regression model, which has been shown to be appropriate in several studies [25,26].
(1)$$y_{t}=\beta_{0}e^{\left(-\beta_{1}e^{-\beta_{2}t}\right)}$$
where $y_{t}$ represents the weight at age t, $\beta_{0}$ represents the asymptotic (mature) weight parameter, $\beta_{1}$ represents the scaling parameter (constant of integration), and $\beta_{2}$ represents the instantaneous growth rate (per day) parameter [27]. SAS 9.4 software’s NLIN procedure was used to estimate parameters (SAS Institute Inc., Cary, NC). The inflection point weight (IPW) and the inflection point age (IPA) are derived using the model parameters as follows:(2)$$IPW=\beta_{0}/e$$
(3)$$IPA=ln\left(\beta_{1}\right)/\beta_{2}$$

The focal sampling technique was used to determine the time budgets for the general behavioral features of feeding, walking, drinking water, scratching, mating, standing, shaking, cleaning, aggressive pecking, wing stretching, lying, and jumping [28]. A digital camera was used for recording the behaviors of six female and six male quails from each experimental group for five minutes on a day of the week in the morning and evening when the birds were four and six weeks old. Three researchers watched from a wide display while the camera recordings were converted into data.

Each week, the durations of tonic immobility (immobilization due to a shocking stimulus) were recorded to determine the fear responses of 10 randomly selected birds from the experimental groups. For the application of the tonic immobility test, the expert operator laid the bird on its back with its head hanging from a particular apparatus. The operator placed one hand gently and without pressure on the bird’s chest and waited 10 s for the bird to become immobile. At the end of 10 s, the operator measured the duration of the bird’s immobility using a stopwatch. The maximum value determined during the inactive time measurement was 300 s [29].

To determine the general amount of stress in quails exposed to various monochromatic lights, developmental stability measurements were conducted. When the quails were three and six weeks old, digital calipers with 0.01 mm precision were used to measure the right and left sides of the face, wing, femur, and fibula, which are bilateral morphological characteristics. The symmetry status of bilateral characteristics (fluctuating, directional, asymmetrical) and relative asymmetry (RA) were determined.
$$RA=100\times \frac{\left|Left+Right\right|}{\left(\frac{Left+Right}{2}\right)}$$

Using the one-sample T-test for identifying the type of symmetry, it was determined if the mean of the group’s differences was significantly different from zero. Similarly, the Shapiro–Wilk test was utilized to establish whether or not the group’s differences were normally distributed. According to the results, the group’s symmetry type was identified [30]. This approach classifies the type of symmetry as fluctuation asymmetry when the distribution is normal and the mean is zero. The form of symmetry that exists when the distribution is normal and the mean is different from zero is directional asymmetry. Antisymmetry is the type of symmetry if the distribution of differences is not normal [31].

When the quails were 42 days old, they were killed. Before being killed, the feed was ceased for four hours. The birds were decapitated using tools, bled out, scalded for two minutes at 55 °C, then manually dissected. After being removed, the fat pad in the abdomen was cooled in an ice-water tank and drained. Each carcass was placed in a plastic bag and kept overnight at +4 °C. The next day, after dissection of the carcasses, the carcasses’ component pieces as well as the leftover abdominal fat on cold carcasses were weighed (in grams) using an electronic digital scale with a 0.01 g sensitivity. Dissection and slaughter were carried out by the same skilled operators. At 42 days old, the percentage of carcass parts yielded were determined in proportion to body weight.

The study employed the profile analysis technique, an element of multivariate analysis of variance (MANOVA), to examine the differentiation between groups with respect to growth samples as measured by weekly individual body weight values. In the statistical analysis of other continuous data obtained in the study, it was determined whether there was a difference between the variance analysis technique and the experimental groups for variables that satisfied the parametric test assumptions at a significance level of 0.05. In the case that there was a statistically significant difference between the groups, the Duncan multiple comparison test was used to determine which group or groups the differences originated from. All statistical analyses utilized the SAS 9.4 statistical software.

## 3. Results

Table 1 presents the results of profile analysis and multivariate analysis of variance (MANOVA), in which weekly individual body weights of quails were analyzed depending on and independently of time. There were statistically significant differences (*p* < 0.05) between the growth profiles of Japanese quails reared under different monochromatic lighting conditions. In the hypothesis testing of the profile analysis’s time intervals, it is determined that the difference in question begins at the age of three weeks and continues until the end of the experiment. However, at the end of experiment, the quails exposed to green, yellow, and blue monochromatic lighting were heavier than the other groups (*p* < 0.05). The parameter estimates and hypothesis test results of the Gompertz model obtained as a result of the nonlinear regression analysis performed using individual growth samples of quails in the experimental groups are presented in Table 2. Concerning the mature weight parameter of the Gompertz growth model for the experimental groups in the study, it was determined that the groups exposed to blue and green monochromatic lighting had higher averages than the other groups (*p* < 0.05). In terms of the other parameters of the Gompertz model and the mean age at the inflection point, there were no statistically significant differences between treatment groups.

While Table 3 presents the findings about the performance and activity behavior characteristics of Japanese quails treated with monochromatic lighting at various wave lengths, Table 4 shows the results of the comfort and social behavior traits of the birds. Among the locomotor behaviors, there was a statistically significant difference between the groups only in terms of lying, and it was determined that the birds exposed to green light had the highest average (*p* < 0.05). On the contrary, the lying time of the quails exposed to yellow monochromatic lighting was found to be lower than the other groups (*p* < 0.05). In the study, the highest averages in terms of aggressive pecking behavior were observed in birds exposed to yellow and red monochromatic light (*p* < 0.05); it is possible to say that birds exposed to green and blue light are less belligerent. In terms of mating behavior, which plays a key role in the social behaviors of poultry, quails exposed to red and yellow monochromatic lighting displayed the highest averages (*p* < 0.05).

The mean values and statistical analysis results of the tonic immobility test used to measure the fear levels of quails exposed to white, green, yellow, blue, and red monochromatic lighting are presented in Table 5. The difference in tonic immobility durations between experimental groups was statistically significant (*p* < 0.05). Accordingly, it was determined that the birds exposed to yellow monochromatic illumination had the highest average tonic immobility. The relative asymmetry averages (%) and statistical analysis results related to bilateral traits of quails in groups treated with monochromatic illumination at different wavelengths are presented in Table 5. There were no statistically significant differences between the treatment groups in terms of the average relative asymmetry of face, wing, femur, and fibula lengths of the quails treated with monochromatic illumination at different wavelengths (*p* > 0.05 for all traits).

Table 6 presents the symmetry types determined by the one-sample T-test and the Shapiro–Wilk normality test of the quails in the experimental groups. According to symmetry types, antisymmetry was identified in three of the four bilateral traits of quails exposed to red monochromatic lighting. In contrast, fluctuating asymmetries were observed in all bilateral characteristics of birds exposed to green monochromatic lighting.

## 4. Discussion

Rozenboim et al. [10] and Cao et al. [32] reported that exposure to monochromatic lighting caused growth disparities at extremely early ages (2 and 4 days old) in broilers. In our study, it is thought that the reason for the emergence of this difference at a later age is due to the fact that the growth rate of quails is not as fast as broiler chickens. Numerous studies indicate that green and blue monochromatic lighting improves growth in birds [12,33,34,35]. In addition, several studies have claimed that the wavelength that reveals the yellow color has positive effects on growth characteristics. Kim et al. [17] reported that the average body weight of chickens treated with yellow monochromatic lighting was higher than those treated with blue and green lighting. Exposing broilers to green, blue, red, and yellow light, Firouzi et al. [36] reported that the highest body weight average at 42 days of age was found in chickens illuminated with yellow. The current study’s findings about the body weight of birds exposed to various monochromatic lighting are compatible with those of Rozenboim et al. [10] and Cao et al. [32] in broiler chickens, and Retes et al. [37] in Japanese quails. The β_0_ parameter estimation averages (248.75–286.99 g) of the Gompertz model for quail growth samples by numerous researchers [27,38,39,40] are compatible with the β_0_ means estimated in this study. The inflection point weights of blue and green groups were also greater (*p* < 0.05), similar to the fact that quails lighted in blue and green had higher average mature weight parameter values.

Huber-Eicher et al. [41], who studied laying hens’ behavior under green, red, and white LEDs, found that the hens tended to feed more under the red and white LEDs than the green ones. According to Mohamed et al. [42], who exposed broilers to blue, green, blue–green mixed, and white monochromatic lighting, hens under blue and green monochromatic lighting were less stressed and more active than those under white light. According to Sultana et al. [43], broiler chickens exposed to varying monochromatic lighting applications exhibited calmer and less active behaviors. Additionally, there was not a significant difference in the behavioral traits of the experimental groups involving feeding and drinking water. In a related study using Pekin ducks, Sultana et al. [44] found that animals exposed to monochromatic lighting of yellow and white were more social and active than animals in other groups. The same research revealed that ducks exposed to blue monochromatic lighting had a higher budget for comfort behaviors like wing stretching than the other groups.

The tonic immobility mean values recorded in Japanese quails exposed to monochromatic lighting at various wavelengths were found to be consistent with the findings (80.8–92.9 s) of Benoff and Siegel [45] and Narinç and Genç [27]. In addition, there are studies in which the tonic immobility duration of Japanese quail is lower (between 26.06 and 53.77 s) [28,46] and higher (between 177.8 and 201.8 s) [47,48,49]. In the current study, the quails with the highest tonic immobility average (100.04 s) were in the yellow monochromatic lighting group, while the tonic immobility mean values of the quails exposed to green and blue monochromatic lighting were the lowest (58.82 and 58.26, respectively). In a study examining the effects of green, blue, white, and yellow monochromatic lighting on the behavior and fear levels of ducks, Sultana et al. [43] found that the tonic immobility durations of ducks exposed to yellow and white lighting were longer than those exposed to blue and green monochromatic lighting. The findings of Sultana et al. [43] were found to be in agreement with our own. In a second study, Sultana et al. [44] found that chickens reared under red LED light had higher tonic immobility durations, although green LED and control (incandescent bulb) groups showed no difference, whereas blue light shortened the tonic immobility duration. The researchers claimed that light with a wavelength between 440 and 570 nanometers would diminish the fear response in chickens. In a similar study, Mohamed et al. [50] found that Mullard ducks that were raised under monochromatic lighting consisting of green and blue light exhibited shorter tonic immobility durations than those that were raised under white and red light. Furthermore, the ducks were less fearful when exposed to green and blue monochromatic lighting. Mohamed et al. [51], who examined the influence of monochromatic lighting on the fear response of broilers, found that birds exposed to green and blue monochromatic lighting had a reduced fear level. The current study’s findings that blue and green monochromatic lighting reduces quail fear are consistent with reports from the literature describing similar research conducted on other poultry species [44,50,51].

Archer [52] and Huth and Archer [53] reported that the relative asymmetry values of broilers reared with two types of LED lighting were not statistically different, whereas conventional lighting increased the relative asymmetry value of chickens. Because directional symmetry and antisymmetry originate from immeasurable sources, fluctuation asymmetry is often used as a primary indicator of the effects of developmental stress factors [54]. However, Grahm et al. [55] claimed that any type of asymmetry can exert the effect of stress. Lens and Van Dongen [56] empirically verified this assumption by demonstrating that when wild birds confront increasing levels of habitat disturbance, there is a transition from fluctuating asymmetry to directional asymmetry. According to Knierim et al. [57], a change in the type of bilateral symmetry may indicate a deterioration in a bird’s ability to cope with stressors throughout its life and is, therefore, a good indicator of animal welfare. Kellner and Alford [58] stated that asymmetry detection may only represent recent growth history and may be effective for evaluating a flock’s current stress level as opposed to its lifetime cumulative stress exposure. In the current study, no directional asymmetry was observed between the symmetry types detected for the lengths of the face, femur, fibula, and wing of quails exposed to monochromatic lighting at different wavelengths.

## 5. Conclusions

In terms of live weight values of Japanese quails, the averages of those treated with blue, green, and yellow light were found to be higher, but according to Gompertz growth curve analysis, it was determined that the maturation capacities of quails treated with blue and green lighting were higher than those of the other trial groups. The best results in terms of both fear and general behavioral characteristics were obtained from quails treated with green and blue monochromatic lighting. A similar situation was observed in bilateral symmetry types, and all symmetry types detected in terms of four bilateral characters of quails subjected to green monochromatic illumination were fluctuating symmetry. As a result, it is thought that the implementation of green or blue monochromatic lighting programs, which provide the best results in terms of growth, behavior, and development balance, can provide economic benefits to producers.

## Figures and Tables

**Table 1 animals-14-00482-t001:** The effect of different wavelengths of light on body weight (g) characteristics.

Time (Day)	White	Green	Yellow	Blue	Red	Intervals (Day)	Profile Analysis
Hatch	8.23	8.21	8.38	8.29	8.46		*p* Value
7	32.31	33.28	32.63	32.77	32.57	0–7	0.768
14	66.15	64.88	67.29	63.68	65.28	7–14	0.125
21	96.80 ^b^	100.03 ^a^	104.22 ^a^	97.60 ^b^	95.77 ^b^	14–21	0.001 *
28	138.53 ^b^	139.34 ^b^	145.17 ^a^	135.10 ^c^	130.24 ^d^	21–28	0.001 *
35	171.28 ^b^	176.00 ^a^	177.06 ^a^	172.12 ^b^	164.65 ^c^	28–35	0.001 *
42	193.33 ^b^	200.68 ^a^	202.28 ^a^	198.24 ^a^	187.36 ^c^	35–42	0.014 *
MANOVA Wilks’ Lambda							0.001 *

MANOVA: Multivariate analysis of variance, * *p* < 0.05, ^a–d^: Different letters within the same line show statistical difference in profile analysis.

**Table 2 animals-14-00482-t002:** The effect of different wavelengths of light on parameters of Gompertz growth model.

Group	β_0_	β_1_	β_2_	IPA	IPW
White	250.86 ^b^	3.12	0.057	20.07	92.29 ^b^
Green	273.32 ^a^	3.25	0.057	20.85	100.55 ^a^
Yellow	258.71 ^b^	3.27	0.062	19.20	95.17 ^b^
Blue	274.99 ^a^	3.24	0.055	21.45	101.16 ^a^
Red	254.52 ^b^	3.23	0.059	19.74	93.63 ^b^
Standard Error	2.20	0.01	0.001	0.13	0.81
*p* Value	0.000 *	0.856	0.485	0.517	0.000 *

* *p* < 0.05, ^a,b^: Different letters within the same column show statistical difference.

**Table 3 animals-14-00482-t003:** The effects of different wavelengths of light on activity behavior characteristics (sec).

Group	Feeding	Drinking	Walking	Lying	Standing	Jumping
White	68.93	9.80	64.13	51.23 ^b^	107.62	1.78
Green	51.13	7.05	55.52	60.50 ^a^	115.58	2.24
Yellow	61.33	8.70	58.18	19.61 ^c^	133.39	1.53
Blue	72.40	10.74	57.42	48.39 ^b^	104.56	3.39
Red	76.19	12.79	55.19	49.15 ^b^	119.73	1.71
Standard Error	6.47	1.92	4.97	8.85	7.62	0.17
*p* Value	0.746	0.890	0.980	0.044 *	0.775	0.134

* *p* < 0.05, ^a–c^: Different letters within the same column show statistical difference.

**Table 4 animals-14-00482-t004:** The effect of different wavelengths of light on comfort and social behavioral characteristics (sec).

Group	Cleaning	Wing Stretching	Shaking	Aggressive Pecking	Scratching	Mating
White	18.24	3.51	3.95	14.44 ^b^	2.78	4.42 ^b^
Green	17.11	4.47	3.50	4.41 ^c^	14.41	3.89 ^c^
Yellow	18.55	3.70	3.78	25.49 ^a^	2.26	6.50 ^a^
Blue	23.13	5.92	4.44	5.92 ^c^	7.48	3.00 ^c^
Red	14.07	2.41	2.75	19.10 ^a^	2.81	6.43 ^a^
Standard Error	2.43	0.53	0.72	1.74	7.15	0.78
*p* Value	0.844	0.367	0.968	0.005 *	0.931	0.018 *

* *p* < 0.05, ^a–c^: Different letters within the same column show statistical difference.

**Table 5 animals-14-00482-t005:** The effect of different wavelengths of light on tonic immobility (sec) and relative asymmetry (%).

Group	Tonic Immobility, sec	Relative Asymmetry, %			
		Face	Wing	Femur	Fibula
White	73.82 ^c^	2.43	1.15	1.57	2.25
Green	58.82 ^d^	1.80	0.37	1.16	0.73
Yellow	100.04 ^a^	1.65	1.17	0.71	7.05
Blue	58.26 ^d^	1.64	1.48	2.74	2.63
Red	86.23 ^b^	1.25	0.77	3.10	3.47
Standard Error	6.88	0.28	0.17	0.28	0.71
*p* Value	0.005 *	0.680	0.304	0.144	0.082

* *p* < 0.05, ^a–d^: Different letters within the same column show statistical difference.

**Table 6 animals-14-00482-t006:** The effect of different wavelengths of light on developmental stability.

Group	Criteria	Face Length	Wing Length	Femur Length	Fibula Length
White	Shapiro–Wilk	0.286	0.320	0.476	0.000
	One-sample T	0.808	0.846	0.715	0.521
	Status	Fluctuation Asymmetry	Fluctuation Asymmetry	Fluctuation Asymmetry	Antisymmetry
Green	Shapiro–Wilk	0.622	0.217	0.483	0.370
	One-sample T	0.210	0.193	0.509	0.087
	Status	Fluctuation Asymmetry	Fluctuation Asymmetry	Fluctuation Asymmetry	Fluctuation Asymmetry
Yellow	Shapiro–Wilk	0.890	0.030	0.813	0.100
	One-sample T	0.877	0.247	0.827	0.793
	Status	Fluctuation Asymmetry	Antisymmetry	Fluctuation Asymmetry	Fluctuation Asymmetry
Blue	Shapiro Wilk	0.002	0.178	0.317	0.038
	One-sample T	0.894	0.205	0.138	0.923
	Status	Antisymmetry	Fluctuation Asymmetry	Fluctuation Asymmetry	Antisymmetry
Red	Shapiro–Wilk	0.000	0.016	0.259	0.000
	One-sample T	0.938	0.280	0.478	0.154
	Status	Antisymmetry	Antisymmetry	Fluctuation Asymmetry	Antisymmetry

## Data Availability

The data that support the findings of this study are available from the corresponding author upon reasonable request.

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
