# Peer review of "The Behavioral and Productive Characteristics of Japanese Quails (Coturnix japonica) Exposed to Different Monochromatic Lighting"

_animals, 2024, doi:10.3390/ani14030482_

Round 1

Reviewer 1 Report

Comments and Suggestions for Authors

At first, this article has a topic that fits the scope of Animals. I think the investigation on the impact of monochromatic lighting for poultry production has great significance, and there is definitely a highlight that this study was on quails. So, I think the novelty and scientific significance have no issues. Regarding methodology, the authors combined several measures to quantify the growth and behavioral characteristics that relate to the influence by monochromatic lighting, which is quite thorough in my opinion. In addition, the statistical analysis is complete and legitimate. The manuscript is overall well-written with very limited writing problems. However, I do have a couple of questions for the authors, which are listed below. Please address both the major and specific questions/comments accordingly.

Author Response

Response to Reviewer 1

First of all, we would like to thank Reviewer 1 for his valuable opinions, suggestions and contributions. We greatly value the suggestions provided in order to improve the manuscript. We extend our sincere appreciation.

1- We also considered using incandescent bulbs, fluorescent or LED lighting as normal lighting for quail, but the environmental factors we used in the experimental design did not allow this. It could be very difficult to prevent light reflection in available treatment environment, especially with traditional lighting equipment. In future studies, we will use traditional lighting as control, especially in studies using different treatment rooms.

2- Discussion section has been shortened.

3- The sentence "The most negative findings on aggressive behavior were observed in birds exposed to monochromatic yellow and red light." was added to the conclusion. (Line 299-301)

4- The phrase "continuous application" " has been added. (Line 26)

5- There are different opsin cells in bird eyes that detect light of different wavelengths. It appears that in quails, opsin cells specialized for wavelengths of blue and green light work better.

6- “The” was deleted. (Line 72)

7- This was the only number of darkrooms we could use in the study. It was not possible to accommodate more cages in the same room due to the reflection of light. (Line 78)

8- Relative asymmetry equation was added. (Line 129)

9- Growth curves are nonlinear regression equations. They are used to make predictions for future observation values using observation values at current time points. Since the early development rate of the yellow group is high and their last week weights are similar, it is expected that the future observation value will be lower than the blue and green groups. (Line 162-165)

Reviewer 2 Report

Comments and Suggestions for Authors

-    In the manuscript animals-2841822 entitled: “The Behavioral and Productive Characteristics of Japanese Quails Exposed to Different Monochromatic Lighting. This study was conducted to determine the effects of white (430-470 nm), green (560 nm), yellow (580 nm), blue (480 nm), and red (660 nm) monochromatic lighting administered over the entire fattening period of Japanese quails on some performance and behavioral characteristics. I find the topic of the manuscript interesting but please state the importance of the topic in the introduction section to show the relevance of the subject in terms of Poultry Science. In my opinion, the manuscript is suitable for publication after major revision.

General comments

-         English grammar and style must be corrected throughout the article. The text is very hard to follow and not clear. There are too many errors to have them specified here, therefore I only provide a general comment to have the English style improved.

-         The abstract is not clear. This important part of the paper simplifies the main findings for a lay readership. Imagine you are trying to describe your work to a random person on the street, this is your challenge and it needs to be much better than this.

-         The introduction section has many relevant data. The introduction should briefly place the study in a broad context and highlight why it is important. It should define the purpose of the work and its significance. The current state of the research field should be carefully reviewed and key publications cited. Please highlight controversial and diverging hypotheses when necessary. Finally, briefly mention the main aim of the work and highlight the principal conclusions.

-         In the discussion section, please be more specific, discuss your study with other similar studies, and state the superiorities of your research compared to previous ones.

-         The conclusions section is too long, please rewrite it to be shorter.

Comments on the Quality of English Language

English grammar and style must be corrected throughout the article. The text is very hard to follow and not clear. There are too many errors to have them specified here, therefore I only provide a general comment to have the English style improved.

Author Response

Response to Reviewer 2

First of all, we would like to thank Reviewer 1 for his valuable opinions, suggestions and contributions. We greatly value the suggestions provided in order to improve the manuscript. We extend our sincere appreciation.

1- In the introduction section, the importance of "illumination and light wavelength" issues for poultry science is stated. (Line 31-44)

2- The article has been proofread from start to finish by a technical reader who is a native English speaker.

3- The abstract is very clear and contains precise results. Unfortunately, in a section with a 200 word limit, there is no space where we can explain our business to people passing by on the street. In a section with such a word limit, you can only present the aim of study, basic material-method, important findings and conclusion in only one sentence.

4- The introduction section is designed to briefly place the study in a broad context and explain why it is important. The purpose of the study is defined in the last paragraph. The current state of the research field has been carefully reviewed and important publications have been cited. In addition, the original value of the subject is also included.

5- Discussion section has been completely revised and made simpler.

6- The conclusion has been summarized and condensed.

Reviewer 3 Report

Comments and Suggestions for Authors

The manuscript no. 2841822 in its current form is unsuitable for publication and requires major corrections. The reviewer's comments are included below.

1. The title of the work should include the Latin name of the Japanese quail in brackets

2. lines 47-49 it is said: „Halevy et al. [13] suggested that the excitation wavelength of blue and green light causes muscular satellite cells (new cells) to proliferate more rapidly, hence accelerating the growth of poultry”. What does (new cells) mean? Satellite cells are stem cells.

3. The authors should provide the correct length of the light waves examined. The Abstract, Introduction and Materials and Methods contain inconsistent data.

Materials and Methods

4. The approval number of the ethics committee should be included in this part of the work.

Authors should describe how animals were killed.

5. lines 88-89: The authors should explain the abbreviations CP and ME and provide the full composition of the used diet, not limiting it only to the cited publication.

6. The authors should provide more details about the LED device used (parameters, origin, company)

7. The authors should clearly define the control group of the study so that the reader does not have to surmise.

8. lines 101-102: The authors should explain the abbreviation NLIN

9. lines 102-106: There is no IPA description of this formula, instead there is IPT (What is IPT?)

10. Why did the authors not evaluate weight differences between males and females using the Gompertz nonlinear regression model?

11. lines 112-114: What was the focal sampling technique? The authors should provide more details. How were the recordings converted into numerical data? On what scale did the authors parameterize the results? Are there any differences between male and female chicks?

12. The authors should explain what tonic immobility means - briefly characterize this term.

13. lines 123-126:  There is no data concerning shank diameters in Table 6.

14. The description of each table should include what statistical test was used to present the data.

15. lines 145-147: „The generalized linear model analysis with Logit function and the Tukey-Kramer test were applied in the statistical analysis of  the qualitative data (mortality rate) acquired during the study”. However, there is no mortality data in this manuscript.

16. Throughout the manuscript there is no information on how the authors presented the obtained data. It is usually stated that the results were presented as mean± SD, SE ,SEM…

17. It would be more transparent for the reader to divide the methodology into different subsections - describing individual analyses. And put a section on statistical analysis at the end.

Results

18. Table 1 - please specify the units in which this data are presented

19. Why is SE not expressed for a particular data in the tables (2,3,4,5)?

20. lines 183-185: „In the study, the highest averages in terms of aggressive pecking behavior were observed in birds exposed to yellow  and green monochromatic light (P<0.05)”.. It should be rather red monochromatic light (see Table 4).

21. lines 186-188: „In terms of mating behavior, which plays a key role in the social behaviors of poultry, quails exposed to red and green monochromatic”…  It should be rather yellow monochromatic lighting (see Table 4).   Authors should be more careful when describing their results.

Discussion

22. lines 255-257: „When the time budgets of the behavioral characteristics of quails exposed to blue and green monochromatic lighting were assessed in the current study, it was revealed that the average locomotor activity were lower”. What type of activity exactly did the authors have in mind? Is it indicated in Table 3?

Conclusion

23. lines 328-330: „In the study, it was determined that the quails in the trial group in which green monochromatic lighting was applied were superior to the other groups in terms of survival”. This statement is unclear because the authors did not provide data on the survival of chicks.

Comments on the Quality of English Language

Language correction recommended

Author Response

Response to Reviewer 3

First of all, we would like to thank Reviewer 1 for his valuable opinions, suggestions and contributions. We greatly value the suggestions provided in order to improve the manuscript. We extend our sincere appreciation.

1- The Latin name (Coturnix japonica) for Japanese quail has been added to the title.

2- The phrase "new cells" has been deleted. (Line 49)

3- Nanometer values for light wavelengths have been updated, thank you very much for your attention.

4- Since the ethics committee number is included at the end of the text in ANIMALS journal, we did not include it in the material and method section. Please see Line 312-313 Institutional Review Board Statement: The animal study protocol was approved by the Akdeniz University's Animal Experiments Local Ethics Committee (03.07.2020/116). The animal slaughter protocol has been added, thank you very much for your attention. (Line 139-147)

“Quails were slaughtered when they were 42 days old. Feed was withdrawn for 4 hours before slaughter. The birds were cut by hand, bled out, scalded (at 55°C for 2 minutes), defeated with equipment, and eviscerated by hand. The abdominal fat pad was taken, chilled in an ice-water tank, and drained. All carcasses were individually placed in plastic bags and stored in +4°C for a night. Next day, after carcass dissection, parts of carcass and the remaining abdominal fat on cold carcasses were weighed (in gram) using an electronic digital balance with a sensitivity of 0.01 g. Slaughter and dissection were performed by same experienced operators. Carcass part yields (%) were calculated in relation to body weight at 42 days old.”

5- The abbreviations CP and ME are written long form. Besides, this study is not an "animal nutrition" branch study, so we did not want to include the feed ration content as an extra table in the already very long article. (Line 90)

6- Company, parameter and origin information has been added for the LED devices used; Mervesan, one chip with 2700 K, Turkey (Line 94)

7- There was no group with any traditional lighting as a control group in the study. Trial groups illuminated with light of different wavelengths are compared with each other. This situation is clearly written in various parts of the article.

8- NLIN is not an abbreviation, it is one of the procedures of the SAS program, it is a name.If it is written as non-linear, it is likely to attract reactions because it is used that way.Sample command codes;

PROC NLIN;

DATA A;

x 1

x 2

...

RUN;

9- Thank you very much for noticing the inflection point time/age confusion. The expressions throughout the article have been changed to "age". (Line 105, 108)

10- The main hypothesis of the study was designed as H0: "the effect of different monochromatic lighting applications on the .... feature". If we were to evaluate the effect of gender on the growth curve, this time we would have to perform two-way analyzes of variance for all traits. This would make the long article even more crowded. We can apply this in our different studies.

11- Surveillance behavior tests are divided into focal and scan sampling. We applied the focal sampling test and the explanations regarding this have been updated. (Line 109-115)

12- Added an explanation for tonic immobility (being immobile due to a shock stimulus) Line 116-117

13- In Line 127, femur and fibula have been added instead of shank, thank you for your attention.

14- The description of the profile analysis was forgotten in the statistical analysis section and has been added. We could not present mortality data in the manuscript because it was requested to be removed because there were too many tables, but we forgot to include their statistical explanations. MANOVA explanation was added in Table 1.Thanks very much for your attention.Since variance analysis was performed in all remaining hypothesis tests, we think that presenting it once in the method section instead of writing "variance analysis technique was used" in the description of each table will prevent repetition.

15- There is no pooled standard error for MANOVA, a multidimensional analysis technique.

All tables except Table 1 contain "standard error of general mean"

16- In Methodology, we divided the paragraphs into sections without subheadings. In addition, the last paragraph explains the statistical analysis.

17- The unit (g) in which the data is presented is added to the table header.

18- The use of standard error of general/pooled means is quite common, it allows readers to see the averages more clearly and the tables are more compact. Since the number of animals in the trial groups is specific in the material section, those who are curious can calculate each one themselves.

19- The sentence was changed to "In the study, the lowest averages in terms of aggressive pecking behavior were observed in birds exposed to yellow and red monochromatic light (P<0.05)….".

20- The sentence was changed to “In terms of mating behavior, which plays a key role in the social behaviors of poultry, quails exposed to red and yellow monochromatic lighting displayed the highest averages (P<0.05).” Thank you for your attention.

21- The sentence “When the time budgets of the behavioral characteristics of quails exposed to blue and green monochromatic lighting were assessed in the current study, it was revealed that the average locomotor activity were lower.” was removed.

22- The conclusion was rewritten and shortened, and findings not included in the article were removed.

Round 2

Reviewer 1 Report

Comments and Suggestions for Authors

The authors have addressed all questions very well. 

Reviewer 2 Report

Comments and Suggestions for Authors

All comments are done as required. Thanks. 

Reviewer 3 Report

Comments and Suggestions for Authors

Comments on the Quality of English Language

English language correction is still needed.